# Mind the Privacy Budget: How Generative Models Spend their Privacy Budgets

## Abstract

Numerous Differentially Private (DP) generative models have been presented that aim to produce synthetic data while minimizing privacy risks. As there is no single model that works well in all settings, empirical analysis is needed to establish and optimize trade-offs vis-à-vis the intended use of the synthetic data. In this paper, we identify and address several challenges in the empirical evaluation of such models. First, we analyze the steps in which different algorithms "spend" their privacy budget. We evaluate the effects on the performance of downstream tasks to identify problem settings they are most likely to be successful at. Then, we experiment with increasingly wider and taller training sets with various features, decreasing privacy budgets, and different DP mechanisms and generative models.

Our empirical evaluation, performed on both graphical and deep generative models, sheds light on the distinctive features of different models/mechanisms that make them well-suited for different settings and tasks. Graphical models distribute the privacy budget horizontally and cannot handle relatively wide datasets, while the performance on the task they were optimized for monotonically increases with more data. Deep generative models spend their budget per iteration, and their behavior is less predictable with varying dataset dimensions, but could perform better if trained on more features. Also, low levels of privacy ($\epsilon \geq 100$) could help some models generalize, achieving better results than without applying DP.

## 1 Introduction

Techniques for creating synthetic data based on generative models have been attracting efforts from the research community (Jordon et al., 2022), government organizations (Benedetto et al., 2018; NIST, 2018a; 2020; NHS England, 2021), regulatory bodies (ICO UK, 2022), and industry alike (TechCrunch, 2022). However, training such models without privacy guarantees can lead to overfitting to the training data or memorization of individual points (Carlini et al., 2019; Webster et al., 2019). This, in turn, enables attacks like membership and property inference (Hayes et al., 2019; Hilprecht et al., 2019; Chen et al., 2020). To mitigate these concerns, models should be trained to satisfy Differential Privacy (DP) (Dwork et al., 2014), whereby noisy/random mechanisms are used to provably minimize the contribution of individual data points to the model.

In this paper, we set out to profile DP generative models for tabular data aiming to understand the settings in which they perform best systematically. As shown by (Hay et al., 2016), the more complex DP algorithms become, the harder it is to analyze their performance analytically. This motivates the need for solid empirical evaluations; thus far, however, evaluations have only been performed on few, relatively small (i.e., involving a dozen features) datasets and rarely beyond a single evaluation metric. Furthermore, it is unclear if/how they scale to larger datasets, even though companies are using scalability delivered in minutes as part of their product offerings (Accelario, 2022; Datagen, 2022; Syntho, 2022; Gretel, 2022).

The conventional wisdom is that computations/methods satisfying DP become more accurate with more training data and less so with stricter privacy guarantees. On the other hand, in some cases, more data or training iterations make deep learning classifiers worse when optimized with DP-SGD (Near & Abuah, 2021). Also, satisfying a small degree of DP improves the performance of CNNs on limited data (Pearce, 2022) and of GANs for imbalanced data (Ganev, 2022), and even defends against reconstruction attacks vs. highly informed adversaries (Balle et al., 2022).

**Problem Statement.** We perform the first empirical evaluation of the effects of different DP mechanisms and the size and dimensionality of training data in the context of generative models and synthetic data. More broadly, we aim to analyze how generative models spend their privacy budget across the number of rows and columns, while also varying the dimensions of the datasets. We also evaluate how the choice of generative model and DP mechanism affects the quality of the synthetic data for downstream tasks, e.g., capturing simple distributions, maintaining high similarity, clustering, and binary and multi-class classification.

We compare two synthetic data approaches, namely, graphical and deep generative models; more precisely, PrivBayes (Zhang et al., 2017) and MST (McKenna et al., 2021b) for the former and DP-WGAN (Alzantot & Srivastava, 2019) and PATE-GAN (Jordon et al., 2018) for the latter.

Overall, we aim to answer the following research questions:

- **RQ1:** How scalable are DP generative models in terms of the dimensions of the dataset?

- **RQ2:** Do DP generative models distribute their privacy budgets in a similar way?

- **RQ3:** What are the effects of different ways to distribute DP and varying dataset dimensions on the downstream task of the synthetic data?

**Main Findings.** Among other things, our experiments reveal that:

1. The graphical models distribute their privacy budget per column, cannot scale to many features (256 for PrivBayes and 128 for MST at most), and increasing the number of rows does not affect the training time. The deep generative models spend their budget per training iteration and can handle much wider datasets but become slower with more data.

2. PrivBayes's performance on downstream tasks degrades when a stricter privacy budget is imposed or the number of features increases, while more data counters these effects. Also, it is the only model properly separating signal from noise in the clustering task.

3. MST is the best performing model at capturing simple statistics and has the best privacy-utility trade-off for this task. Also, when there is a lack of data, it benefits from adding a small degree of privacy ($\epsilon \geq 100$). Increasing the number of rows too much, however, can cause MST to overfit and degrade its performance on more complex tasks.

4. The GAN models have more variable behaviors with different dataset dimensions. While they are not as competitive on simple tasks and frequently cannot beat baseline models, PATE-GAN is well-suited for more complex tasks and often outperforms both graphical models. PATE-GAN can also improve when presented with more features and is better than DP-WGAN for almost all settings.

Overall, we are confident that our work will assist researchers and practitioners deploying DP synthetic data techniques in understanding the trade-offs and navigating through the best candidate models, vis-à-vis the dataset features, desired privacy level, and the downstream task at hand.

## 2 Background and Related Work

In this section, we review related work on Differential Privacy and data dimensionality for queries, classification models, and generative models. In Appendix A, we also provide background information on DP, synthetic data generation, and DP generative models.

**Notation.** In the rest of the paper, we use $n$ to denote the number of rows and $d$ and the number of columns of the real dataset. Also, $\epsilon$ denotes the privacy budget and $\delta$ the probability of failure in a DP mechanism (see Equation 1 in Appendix A).

**DP Queries and Data Dimensionality.** (Hay et al., 2016) benchmark 15 DP algorithms for range queries over 1 and 2-dimensional datasets, showing that increasing values of $n$ reduce error. For small $n$, data-dependent algorithms tend to perform better; for large $n$, data-independent algorithms dominate. For (more complex) predicate counting queries and higher dimensional data, (McKenna et al., 2021a) propose a method with consistent utility improvements and show that increasing $d$ results in more significant errors. However, they experiment with datasets with at most 15 features, and their model struggles to scale beyond 30-dimensional datasets.

| DP Model | Model Type | DP Mechanism | Max $d$ |
|---|---|---|---|
| Independent | Marginal | Laplace | 1,024 |
| PrivBayes (Zhang et al., 2017) | Graphical | Exponential + Laplace | 256 |
| MST (McKenna et al., 2021b) | Graphical | Exponential + Gaussian | 128 |
| DP-WGAN (Alzantot & Srivastava, 2019) | Deep Learning | DP-SGD | 1,024 |
| PATE-GAN (Jordon et al., 2018) | Deep Learning | PATE | 1,024 |

**Table 1:** DP generative models used in this paper.

**DP Classifiers and Data Dimensionality.** For standard predictive models, more data and longer training usually lead to better performance, i.e., lower loss/better classification on the training set. This also holds for DP logistic regression learned via empirical risk minimization and objective perturbation (Chaudhuri et al., 2011): for large $n$, the cost function tends to the non-private one, and while the scale of the noise is independent of $d$, there is a linear shift to the objective function that does not affect the optimization if it is strongly peaked (Antonova, 2016). However, this does not necessarily hold for other DP models; e.g., methods based on iterative training like DP-SGD trade-off the number of training steps and the noise added per iteration (Near & Abuah, 2021).

**DP Generative Models and Data Dimensionality.** Unlike query answering and classification tasks, the output of generative models lives in a high-dimensional space. Thus, it has much higher sensitivity, which even trying to analyze is an incredibly difficult task. Furthermore, private synthetic data generation is computationally challenging (exponential in $d$ in the worst-case scenario, i.e., all two-way marginals are preserved (Dwork et al., 2009; Ullman & Vadhan, 2011)). Nevertheless, these results are in the worst-case and do not rule out the existence of practical algorithms (such as those introduced in Appendix A); indeed, if most, rather than all, correlations are preserved, one can build computationally efficient algorithms (Boedihardjo et al., 2021).

Since there is no "one-size-fits-all" DP synthetic data generation method, (Jordon et al., 2022) highlight the need to empirically assess the privacy and fidelity of the data on a *per-case* basis. Even though (Hay et al., 2016) describe a set of standardized evaluation principles for DP query answering algorithms, including diversity of inputs such as varying domain size, scale, and shape, no such studies focus on synthetic data generation. For example, current frameworks for synthetic data evaluation (Arnold & Neunhoeffer, 2020) do not consider varying $n$ and $d$ as essential factors, and benchmark studies (Tao et al., 2022) do not consider datasets with more than 41 features.

To our knowledge, state-of-the-art graphical models have not been evaluated on datasets with more than 100 dimensions. Specifically, for PrivBayes and MST, (Takagi et al., 2021) argue that the former only performs well for datasets having simple dependencies and a small number of features, while the latter cannot reconstruct the essential information from limited information (i.e., 1 and 2-way marginals) required for more complex classification tasks. Finally, (Li et al., 2022a) observe that the two models are usually tested on tabular datasets with dozens of dimensions and claim that they still suffer from the curse of dimensionality.

Overall, our work aims to bridge these gaps through an extensive empirical evaluation, testing both graphical models and deep generative models on diverse dataset sizes and shapes as well as a variety of downstream tasks with different complexity.

## 3 PRELIMINARIES

In this section, we introduce the datasets and the DP generative models we use in our evaluation and discuss the specific steps in which they distribute their privacy budgets.

### 3.1 GENERATIVE MODELS

Table 1 lists the DP generative models used in our analysis. The specific implementations we use are given in footnotes. (Note that we independently implemented Independent and PrivBayes.) For all experiments, the default hyperparameters are used unless stated otherwise. We focus on two types of generative approaches, graphical models (PrivBayes and MST) and GANs (DP-WGAN and PATE-GAN). The former model the joint distribution by breaking it down to explicit lower-dimensional marginals. The latter approximate the distribution implicitly by iteratively optimizing two competing neural networks until reaching an equilibrium: a generator, creating synthetic data

from noise, and a discriminator, separating real from synthetic data points. We select these models as they are well-studied and tested, have reliable open source implementations, and have proven to be among the top performing solutions in practice, winning both the Differential Privacy Synthetic Data Challenge (NIST, 2018a) and The Unlinkable Data Challenge (NIST, 2018b) NIST competitions. Also, they rely on different modeling techniques and DP mechanisms.

**Independent.** As a baseline, we use a simple model that, for all columns, independently captures noisy marginal counts through the Laplace mechanism and then samples from them to generate synthetic data. Though very simple, it has proven to perform better than far more sophisticated models in certain settings (Tao et al., 2022).

**PrivBayes (Zhang et al., 2017)** first constructs an optimal Bayesian network by randomly picking up a node (column of the dataset) as the root and iteratively adding one node at a time, maximizing the mutual information between the existing "parent" nodes and a candidate "child" node. Half of $\epsilon$ is spent at this step using the Exponential mechanism, while the network degree argument determines the maximum number of parents a given node can have. Then, PrivBayes estimates the resulting low-dimensional conditional distributions through computing noisy contingency tables (utilizing the Laplace mechanism) before normalizing and converting them to distributions. Since the network is built iteratively and all computed distributions are conditional, they are also consistent. We run PrivBayes for datasets with columns up to 256, setting the network degree to the default 3 for datasets with fewer than 100 columns and to 2 otherwise.

**MST (McKenna et al., 2021b)**[1] follows a similar procedure. For selection, it starts with all 1-way marginals and finds attribute pairs (2-way marginals) that form a maximum spanning tree of the underlying correlation graph. This is achieved through first estimating all 2-way marginals using Private-PGM (McKenna et al., 2019) (a post-processing method that infers a data distribution given noisy measurements) and, then, one by one, noisily adding a highly weighted edge to the graph. The edge weights are measured by the $L_1$ distance between real and estimated 2-way marginals. All selected marginals are measured privately using the Gaussian mechanism. MST allocates $1/3$ of the privacy budget to selection and the remaining $2/3$ to measurement. We train MST on datasets with up to 128 columns and set $\delta = 10^{-5}$ (to be consistent with DP-WGAN and PATE-GAN).

**DP-WGAN (Alzantot & Srivastava, 2019)**[2] utilizes the Wasserstein GAN (WGAN) architecture (Arjovsky et al., 2017), which achieves better learning stability and improves mode collapse issues compared to the original GAN by using Wasserstein distance rather than Jensen-Shannon divergence. The model relies on DP-SGD to ensure the privacy of the discriminator during training, which in turn guarantees the privacy of the generator since it is never exposed to the real data.

**PATE-GAN (Jordon et al., 2018)**[3] adapts the PATE framework and combines it with a GAN architecture. The architecture consists of a single generator, $t$ teacher-discriminators, and a student-discriminator. The teacher-discriminators are only presented with disjoint subsets of the training data and are trained to improve their loss with respect to the generator (i.e., classifying samples as real or fake). In contrast, the student-discriminator is trained on noisily aggregated labels provided by the teachers, and its loss gradients are sent to train the generator.

## 3.2 PRIVACY ANALYSIS

The two types of models substantially differ from each other from a DP perspective, e.g., what DP mechanisms they use, how they distribute their privacy budgets, what factors cause more considerable expenditures, etc. The graphical models rely on the "select-measure-generate" paradigm, i.e., the first two steps are: 1) selecting a collection of low-dimensional marginals and 2) measuring them with a noise addition mechanism. Naturally, as $d$ increases, the privacy budget needs to be distributed among more marginals, potentially leading to more noisy measurements. By increasing $n$, however, we could expect to decrease the per-measurement sensitivity and thus resulting in more accurate estimations.

---

[1]https://github.com/ryan112358/private-pgm
[2]https://github.com/nesl/nist_differential_privacy_synthetic_data_challenge
[3]https://bitbucket.org/mvdschaar/mlforhealthlabpub

| Dataset | Max $n$ | Max $d$ | Downstream Task Evaluated |
|---|---|---|---|
| *Eye Gauss* | 128k | 1,024 | Statistics: Mean (Fig. 4) and Correlation (Fig. 5), PCA (Fig. 12, 13) |
| *Corr Gauss* | 128k | 1,024 | Statistics: Mean (Fig. 6) and Correlation (Fig. 1, 7), PCA (Fig. 14, 15) |
| *Mix Gauss Unsup* | 128k | 1,024 | PCA (Fig. 16, 17), Clustering (Fig. 11) |
| *Mix Gauss Sup* | 128k | 1,024 | PCA (Fig. 18, 19), Classification (Fig. 20) |
| *Census* | 199k | 41 | Similarity: Marginal and Mutual Info (Fig. 2, 8, 10), Classification (Fig. 3a, 21) |
| *MNIST* | 60k | 784 | Classification (Fig. 3b, 3c) |

**Table 2:** Datasets and evaluated downstream tasks.

As for GANs, one of the most widely used approaches is to train them iteratively using mini-batches based on DP-SGD. For fixed-networks architectures, increasing $d$ should not be a major factor as that only affects the discriminator's input and the generator's output layers. Analyzing the effect of increasing $n$ is more complicated: on the one hand, more (clean and diverse) training data is usually better for the model as that helps it generalize. On the other hand, more iterations could make the model worse since one has to use a lower privacy budget per iteration, and so the scale of the noise goes up (Near & Abuah, 2021).

**Independent.** Unlike the other four models, this is the only one that distributes its DP budget independently of the data. It always selects the same marginals (all columns) and uses the same amount of budget to get every noisy marginal, $\epsilon/d$.

**PrivBayes and MST.** In total, PrivBayes measures approx. $d$ 4-dimensional (if the network degree is three) noisy distributions, each of which has allocated $\epsilon/2d$. The MST measurement step has a few advantages over PrivBayes: i) it devotes more budget to it (2/3 vs. 1/2); ii) it models structural zeros in the distribution (areas with negligible probability); iii) it uses the Gaussian mechanism, which has better bounds than Laplace; and iv) even though in total it measures more marginals (approx. $2d$ 1 or 2-way vs. $d$ 4-way), their dimensionality is lower, which means that the noise could be distributed more efficiently. However, it requires more computations, potentially leading to slower running times as Private-PGM is called twice, namely, in the selection and generation steps.

**DP-WGAN and PATE-GAN.** In DP-WGAN, the privacy budget is not directly computed but estimated using the moments accountant method (Abadi et al., 2016). Unlike graphical models, DP-WGAN spends its privacy budget iteratively, through DP-SGD. PATE-GAN also relies on the moments accountant. An advantage over DP-WGAN is that noise is not added directly to the gradients but to the vote of the teacher-discriminators. Furthermore, the accountant in PATE-GAN would attribute a lower privacy cost to accessing noisy aggregations (from the teacher-discriminators) with stronger consensus as a single teacher/data point would have a lower influence on the final vote.

## 3.3 DATASETS

For our experiments, we create four progressively more difficult controllable datasets based on the normal distribution (*Eye Gauss*, *Corr Gauss*, *Mix Gauss Unsup*, *Mix Gauss Sup*) to test whether a generative model can capture different aspects of the distribution. We also include two standard, commonly used datasets, namely, *Census* and *MNIST*, aiming to be consistent with other studies (Tao et al., 2022; Pearce, 2022; Ganev, 2022). A summary of the datasets is reported in Table 2, and more detailed descriptions, along with the dimension ranges, can be found in Appendix B.

## 4 EXPERIMENTAL EVALUATION

In this section, we present and analyze our experiments with four DP generative models (plus a baseline model) and six datasets over several underlying tasks – scalability, statistics, similarity, and classification. Due to space limitations, the evaluation of Principal Component Analysis (PCA) on all *Gauss* datasets, as well as of the clustering task, are presented in Appendix C.4.

For all generative models and all privacy budgets, we train the model $m = 5$ times and generate $s = 5$ synthetic dataset, which yields 25 synthetic datasets for each reported point in the plots. Besides varying the dataset dimensions (as discussed in Sec. 3.3), we also vary the privacy budget $\epsilon$ in the range $\{0.01, 0.1, 1, 10, 100, 1k, 10k, \infty\}$. In total, we train 15k generative models and generate 75k synthetic datasets. A summary of all our experiments is reported in Table 2.

| DP Model↓ $d\rightarrow$ | 8 | 16 | 32 | 64 | 128 | 256 | 512 | 1,024 |
|---|---|---|---|---|---|---|---|---|
| Independent | 0.00 | 0.01 | 0.01 | 0.03 | 0.08 | 0.22 | 0.89 | 3.03 |
| PrivBayes | 0.01 | 0.02 | 0.06 | 0.26 | 2.29 | 34.15 | | |
| MST | 0.75 | 1.55 | 3.23 | 7.71 | 35.90 | | | |
| DP-WGAN | 0.50 | 0.58 | 0.76 | 1.24 | 3.53 | 6.47 | 12.52 | 23.97 |
| PATE-GAN | 0.65 | 0.68 | 0.76 | 0.94 | 1.40 | 2.13 | 4.09 | 10.38 |

**Table 3:** Running time (in minutes) of the model's fitting step for DP generative models, on *Corr Gauss*, varying $d$ and $n = 16k$.

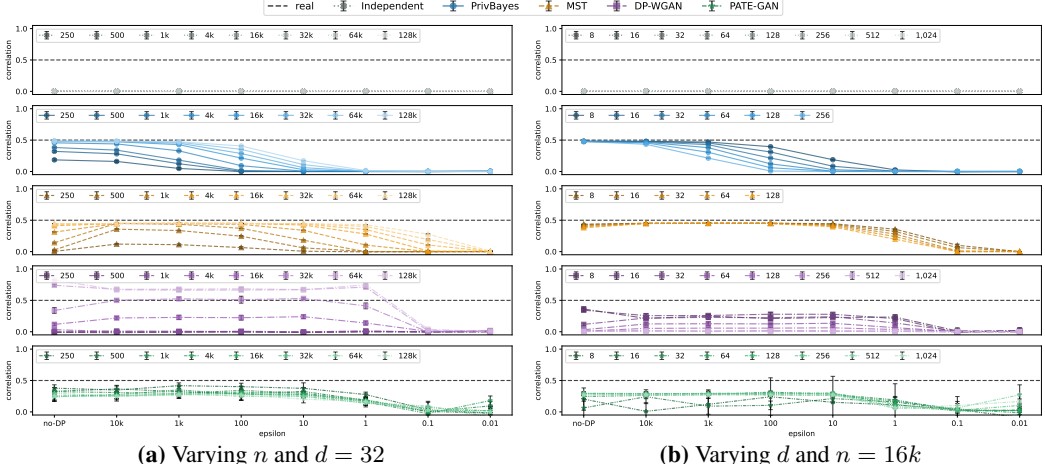

(a) Varying $n$ and $d = 32$         (b) Varying $d$ and $n = 16k$

**Figure 1:** Off-diagonal pairwise correlation for different $\epsilon$ levels, on *Corr Gauss*, varying $n$ and $d$.

## 4.1 SCALABILITY

We show the running time of the models' fitting step on *Corr Gauss* with varying $d$ in Table 3. In Appendix C.1, we also report the running times of the fitting and generation (for a fitted model) steps, while varying $n$ and $d$ (see Tables 4, 5, 6). All experiments are run on an AWS instance (m4.4xlarge) equipped with a 2.4GHz Intel Xeon E5-2676 v3 (Haswell) processor, 16 vCPUs, and 64GB RAM. We opt to set very practical time constraints to stress test scalability *claims* from commercial players (Accelario, 2022; Datagen, 2022; Syntho, 2022; Gretel, 2022); specifically, we discard models taking longer than 60 minutes to train. (This also allows us to maintain a reasonable amount of models to test – i.e., 15k trained model instances.)

The graphical models scale polynomially with $d$ and quickly approach the 60 minutes limit — it takes PrivBayes 35 minutes to fit on a 256-dimensional dataset and MST 36 minutes on 128 dimensions while increasing $n$ is barely a factor. On the other hand, increasing either of $d$ or $n$ slows the GAN models — $d$ due to the increase of the first and last layers of, respectively, the discriminator and generator and $n$, due to the increased number of iterations (we fix the number of epochs instead). PATE-GAN is far more time efficient than DP-WGAN as PATE only adds noise to the teacher-discriminator votes, while DP-SGD clips and adds noise to all discriminator layers. For all models apart from Independent, the generation step only takes a fraction of the training time.

## 4.2 STATISTICS

We report the off-diagonal pairwise correlation across all columns of *Corr Gauss* with varying $n$ and $d$ in Fig. 1. We visualize the remaining statistics for *Eye Gauss* and *Corr Gauss* in Appendix C.2 (see Fig. 4, 6 for mean and Fig. 5, 7 for other pairwise correlation).

Overall, MST captures the distributions best, especially the mean and other pairwise correlation, and it is the least sensitive to changing $n$ and $d$. Looking at the off-diagonal pairwise correlation in Fig. 1, we observe that, for $n \leq 4k$, the correlation score *improves* when privacy is applied and always remains above the "no-DP" values for $\epsilon \geq 10$. In other words, DP acts as regularization when there is insufficient data for the model to capture the distribution. To the best of our knowledge, this is the first such finding of non-deep learning DP model as previous examples, CNNs (Pearce,

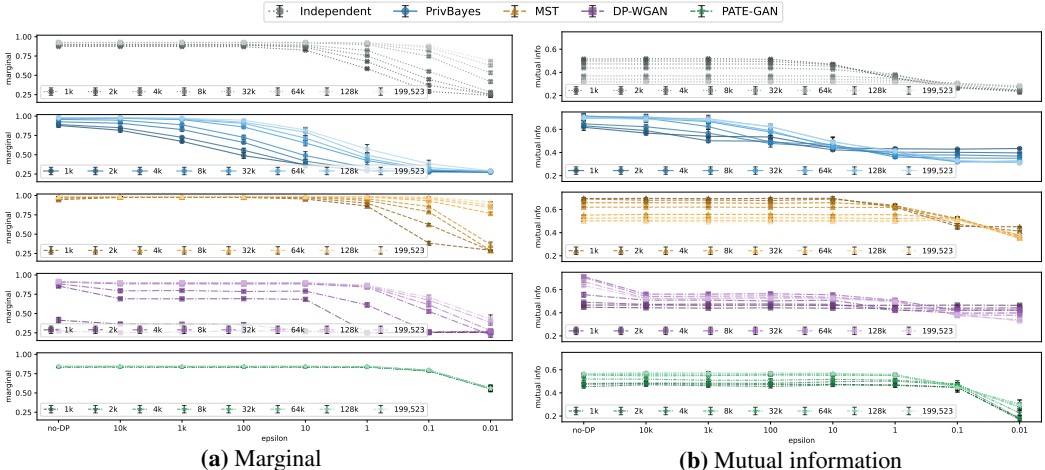

**(a)** Marginal            **(b)** Mutual information

**Figure 2:** Marginal and pairwise mutual info similarity for different $\epsilon$ levels, on *Census*, varying $n$.

2022) and GANs (Ganev, 2022), are neural networks relying on, respectively, DP-SGD and PATE. However, MST cannot scale beyond 128 features.

PrivBayes can handle wider datasets, up to 256 columns. It performs closest to what one would expect, as mean and other pairwise correlation stay close to 0 for all $n$ and $d$, for both datasets. However, for off-diagonal pairwise correlation (see Fig. 1), there is a monotonic improvement with increasing $n$ and deterioration with increasing $d$, reaching the baseline levels of Independent for different levels of $\epsilon$.

The GAN models behave less predictably, and their performance is not monotonic when the dimensions are varied. In all cases, PATE-GAN outperforms DP-WGAN, almost on par with MST for $\epsilon > 0.1$ for *Eye Gauss*. For *Corr Gauss*, however, it cannot capture the off-diagonal pairwise correlation sufficiently well, creating data with correlation closer to 0.3 as opposed to 0.5 for the real one. Interestingly, varying the dataset dimensions affects DP-WGAN differently — for both datasets, increasing $n$ yields more correlated distributions with mean around 0, while, for *Corr Gauss*, increasing $d$ (beyond 128) makes the model generate data with relatively uncorrelated columns ($\leq 0.2$) but with mean further away from 0 ($> 0.5$). In fact, for *Corr Gauss*, the model cannot distinguish between the off-diagonal and other correlations for $d = 32$ and creates data with equal correlation.

## 4.3 SIMILARITY

Next, in Fig. 2, we plot marginal and pairwise mutual information similarities between the real and synthetic datasets for *Census* with varying $n$. In Appendix C.3, we also report a zoomed-in plot for MST and PATE-GAN (see Fig. 8), visualize fitted PrivBayes and MST networks (Fig. 9), and break down the pairwise mutual information for connected vs. non-connected nodes in (Fig. 10).

Looking at the marginal similarity, Independent, PrivBayes, MST, and DP-WGAN behave as expected, as higher $n$ results in monotonically better scores. The fact that Independent outperforms PrivBayes and the GANs for $\epsilon \leq 10$ should not surprise as it only captures the marginal distributions and does not "waste" any privacy budget for other purposes. With increasing $n$, there is also an improvement for PATE-GAN, but the sensitivity is much lower. Fig. 8a shows that, when there is only little data available ($n < 4k$), adding some privacy ($10k \leq \epsilon \leq 10$), once again, helps MST.

As for the mutual information similarity (see Fig. 2b), adding more data points to the training data yields worse scores for Independent and MST. For the latter, this could be due to the overfitting of the model to its objective function (i.e., all 1-way and 41 2-way marginals as seen in Fig. 9b), thus failing to capture all 2-way marginals. This is also visible in Fig. 10, where MST suffers a more significant drop in scores of connected vs. non-connected nodes compared to PrivBayes (which in turn models a much larger number of connections, 120, as seen in Fig. 9a). For $n > 8k$, PATE-GAN outperforms MST (Fig. 8b); this could be explained by its training procedure, which prioritizes creating plausible synthetic data points (i.e., implicitly maintaining correlations between columns).

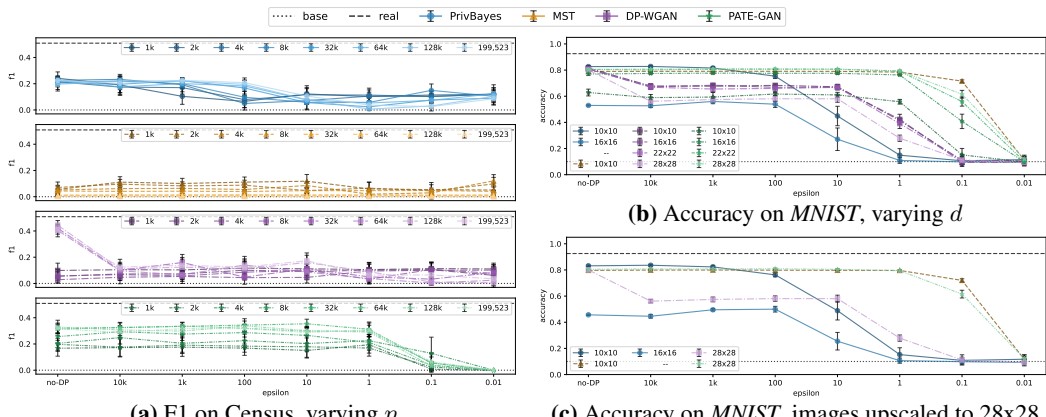

**(b)** Accuracy on *MNIST*, varying $d$

**(a)** F1 on Census, varying $n$  **(c)** Accuracy on *MNIST*, images upscaled to 28x28

**Figure 3:** F1 and accuracy for different $\epsilon$ levels, on *Census*, varying $n$ and *MNIST*, varying $d$.

Finally, with PrivBayes and DP-WGAN, we observe that increasing $n$ helps the score for higher levels of privacy, but only up to $\epsilon = 1$. Beyond this, one would be better off training the models on less training data.

## 4.4 CLASSIFICATION

Finally, we report the F1-score on *Census* with varying $n$ and accuracy on *MNIST* with different resolutions (features) in Fig. 3a and 3b. In Appendix C.5, we present additional classification results, i.e., accuracy on *Mix Gauss Sup* with varying $n$ and $d$ (see Fig. 20), and accuracy (along with F1) on *Census* (Fig. 21).

Looking at *Mix Gauss Sup*, PrivBayes, once again, behaves as expected for different $n$ and $d$. There is a common trend for MST and DP-WGAN, as both models need at least 16k data points achieve better than random accuracy. Similarly, if there are too many features, 128 for MST and $\geq$256 for DP-WGAN, the accuracy approaches the random baseline. Varying $n$ does not seem to be a significant factor for PATE-GAN as accuracy tends to be close to the real one for $\epsilon > 0.1$. Increasing $d$, however, has a negative effect, which is in contrast with previous experiments. As for *Census*, we consider both accuracy and F1 as the dataset is imbalanced (93.8% of the people make less than 50k/year). Somewhat unexpectedly, DP-WGAN comes closest to the real F1 baseline but only for the "no-DP" case. Overall, PATE-GAN is the only model with a F1 score consistently close to 0.35, provided that it was trained on at least 8k points. For MST, increasing $n$ helps accuracy (see Fig. 21) but at the expense of F1, meaning that the classifiers trained on the synthetic data overfit to the majority class.

These trends are very close to the one observed in the similarity experiments from Sec. 4.3; the relative overperformance of the GAN models compared to MST is in direct contradiction with previous studies (Tao et al., 2022). Although we cannot say for sure, we believe that this might, in part, be due to not using the original GAN implementations but relying on third-party ones. Unfortunately, (Tao et al., 2022) dismiss them as good candidate models while arguably overstating the capabilities of other approaches.

As for the accuracy on *MNIST* (Fig. 3b), we see that more features (i.e., images with higher resolution/better quality) deteriorate the performance of PrivBayes and DP-WGAN but help PATE-GAN. In principle, we could expect higher-resolution images to improve both GAN models, but this is the case only for PATE-GAN. However, none of the models approach the real baseline (0.9 accuracy). While this is expected for graphical models, it is somewhat surprising for the GANs. We believe this is because both GANs rely only on feed-forward layers and not CNNs. MST trained on 10x10 images performs very well, achieving 0.8 accuracy, on par with PATE-GAN ($d > 10$x10) for $\epsilon > 0.1$. Again, this might be surprising, as MST does not explicitly model higher-level dependencies, which is important in the image domain.

## 4.5 TAKE-AWAYS

Overall, our experiments allow us to answer our three research questions introduced in Sec. 1.

*RQ1: How scalable are the DP generative models in terms of the dimensions of the dataset?* Graphical models suffer from the curse of dimensionality, and PrivBayes and MST scale up to 256 and 128 columns, respectively (under practical computation restrictions). The GAN models scale to much wider datasets but become slower when presented with taller datasets (since we fix the number of training epochs).

*RQ2: Do DP generative models distribute their privacy budgets in a similar way?* No. The graphical models distribute their privacy budget horizontally and the GANs vertically (i.e., they spend their budget per iteration).

*RQ3: What are the effects of different ways to distribute DP and varying real dataset dimensions on the downstream task of the synthetic data?* The effects are mixed. Overall, more training data helps the graphical models but sometimes causes MST to overfit and degrade its performance on tasks that require capturing complex relationships (e.g., classification). Also, in some instances, a small degree of privacy helps MST when presented with a limited amount of data. Varying the dataset dimensions is more unpredictable (more variable and usually not monotonic) for the GAN models. While they underperform at simple tasks on controllable datasets, PATE-GAN could be very competitive at more challenging tasks and could improve if the dataset is higher dimensional.

## 5 DISCUSSION AND CONCLUSION

In this paper, we studied and compared how different DP generative models distribute their privacy budget, how they scale in terms of increasing dimensions of the dataset, and how these factors affect the quality of the resulting synthetic data for a variety of tasks. We did so to shed light on which models are best suited for different settings and tasks. We showed that graphical models work best with simpler tasks and lower-dimensional datasets, while deep generative models on more complex problems and wider datasets.

In the rest of this section, we discuss potential first-cut solutions and future research directions.

**Increasing Number of Rows.** From our experiments, we observed that more data does not always translate to improved quality for all models and evaluations (e.g., the GAN models and MST, apart from marginal similarity). On the other hand, for some models (MST and DP-WGAN), a minimum data threshold is needed to perform better than random. Therefore, more research is needed to find a good balance. One avenue could be to investigate optimal times for early stopping or dataset sampling techniques. Also, one could build relevant public datasets for the tabular domain and develop pre-trained models; researchers and practitioners could then fine-tune the models on their specific (private) dataset (Harder et al., 2022). This approach has proved to be very promising in other areas, including NLP (Li et al., 2022b; Yu et al., 2022) and vision (Tramer & Boneh, 2021; Golatkar et al., 2022; De et al., 2022).

**Increasing Number of Features.** Our evaluation also showed that increasing the data features results in synthetic data with progressively worse performance on the downstream task (except for PATE-GAN with *MNIST*). Furthermore, the graphical models (which perform better at lower dimensional datasets) cannot scale beyond 128/256 dimensions. A logical follow-up would be to try and reduce the dimensionality of the dataset, train/generate synthetic data in the lower space (this would also help with the DP budget) using the better-suited models, and, if necessary upscale to the original space. (Tantipongpipat et al., 2021) propose a similar approach, combining VAE and GAN.

As a proof of concept, in Fig. 3c, we downscale *MNIST* images using standard tools to 10x10/16x16, train MST (on 10x10) and PrivBayes (on 10x10 and 16x16), generate synthetic images, and then upscale them back to 28x28. Comparing classifiers trained on both real and synthetic images, we observe that: 1) classifiers trained on MST generated data perform very well (on par with PATE-GAN trained on the original images), and 2) neither of MST/PrivBayes lose utility compared to when they were tested on downscaled real images (Fig. 3b). Another way to improve on a particular task would be to investigate the most relevant/important features and spend the privacy budget strategically (Rosenblatt et al., 2022). Alternatively, we could choose the simplest "good enough" model at hand, e.g., Independent preserved marginal similarity even in high dimensions.

Overall, our work takes an important step in profiling state-of-the-art approaches for DP synthetic data generation, and their use for downstream tasks, and paves the way for further research and improvements.

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

## A  ADDITIONAL BACKGROUND AND RELATED WORK

**Differential Privacy (DP).** A randomized algorithm $\mathcal{A}$ satisfies $(\epsilon, \delta)$-DP if, for all of its possible outputs $S$, and all neighboring datasets $D$ and $D'$ ($D$ and $D'$ are identical except for a single data row), the following holds (Dwork et al., 2006b; 2014):

$$P[\mathcal{A}(D) \in S] \leq \exp(\epsilon) \cdot P[\mathcal{A}(D') \in S] + \delta \tag{1}$$

Here $\epsilon$ is a positive, real number (also referred to as the *privacy budget*) that quantifies the information leakage, while $\delta$, usually a very small real number, allows for a probability of failure. In other words, looking at the output of the computation (e.g., the trained model), it is impossible to distinguish whether or notany individual's data was included in the input dataset.

In this paper, we study generative models relying on a variety of well-known DP techniques: the Laplace (Dwork et al., 2006b), Exponential (Dwork et al., 2006a), and Gaussian mechanisms (McSherry & Talwar, 2007), Differentially Private Stochastic Gradient Descend(DP-SGD) (Abadi et al., 2016), and Private Aggregation of Teacher Ensembles (PATE) (Papernot et al., 2017; 2018).

Due to its composition and robustness to post-processing properties, DP allows different DP mechanisms to be combined while the overall privacy budget is tracked and DP-trained models to be re-used without further privacy leakage.

**Synthetic Data.** In this paper, we focus on techniques to create synthetic data through generative machine learning models. A sample dataset $D^n$, consisting of $n$ iid drawn records with $d$ features from the population $D^n \sim P(\mathbb{D})$, is fed as input to the generative model training algorithm $GM(D^n)$ during the fitting step. In turn, $GM(D^n)$ updates its internal parameters $\theta$ to learn $P_{\bar{\theta}}(D^n)$, (a lower-dimensional) representation of the probability of the sample dataset $P(D^n)$, and outputs the trained model $\overline{GM}(D^n)$. Then, $\overline{GM}(D^n)$ could be sampled to generate a synthetic data of size $n'$, $S^{n'} \sim P_{\bar{\theta}}(D^n)$. Since both the fitting and generation steps are stochastic, one can train the generative model $m$ times and sample $s$ synthetic datasets for each trained model to get confidence intervals.

**DP Generative Models.** There is a vast literature of DP techniques for tabular synthetic data generation, namely, copulas (Li et al., 2014; Asghar et al., 2021; Gambs et al., 2021), graphical models (Zhang et al., 2017; 2019; McKenna et al., 2021b; Cai et al., 2021; Mahiou et al., 2022), workload/query based (Vietri et al., 2020; Aydore et al., 2021; Liu et al., 2021), Variational Autoencoders (VAEs) (Acs et al., 2018; Abay et al., 2018; Takagi et al., 2021), Generative Adversarial Networks (GANs) (Xie et al., 2018; Zhang et al., 2018; Jordon et al., 2018; Alzantot & Srivastava, 2019; Frigerio et al., 2019; Tantipongpipat et al., 2021; Long et al., 2021), and other approaches (Chanyaswad et al., 2019; Zhang et al., 2021; Ge et al., 2021).

## B  DATASETS

**Gaussians.** We create four progressively more complex datasets based on the gaussian distribution:

- *Eye Gauss* consists of columns that are independently distributed standard normals.
- *Corr Gauss* (inspired by (Belghazi et al., 2018; Poole et al., 2019; Rhodes et al., 2020)) is a multivariate normal with mean 0 and covariance matrix with 1s on the diagonal (unit variance), 0.5s on the off-diagonal (all neighboring columns have correlation 0.5), and 0s everywhere else (not neighboring columns are uncorrelated). The idea is to see if the model would be able to capture the correlation correctly.

| DP Model↓ $n \rightarrow$ | 250 | 500 | 1k | 4k | 16k | 32k | 64k | 128k |
|---|---|---|---|---|---|---|---|---|
| Independent | 0.01 | 0.01 | 0.01 | 0.01 | 0.01 | 0.02 | 0.03 | 0.05 |
| PrivBayes | 0.02 | 0.03 | 0.03 | 0.05 | 0.06 | 0.07 | 0.10 | 0.13 |
| MST | 3.23 | 3.27 | 3.27 | 3.28 | 3.23 | 3.23 | 3.32 | 3.30 |
| DP-WGAN | 0.11 | 0.11 | 0.13 | 0.24 | 0.76 | 1.42 | 2.73 | 5.37 |
| PATE-GAN | 0.02 | 0.03 | 0.05 | 0.18 | 0.76 | 1.75 | 4.55 | 13.40 |

**Table 4:** Running time (in minutes) of the model's fitting step for DP generative models, on *Corr Gauss*, varying $n$ and $d = 32$.

| DP Model↓ $n \rightarrow$ | 250 | 500 | 1k | 4k | 16k | 32k | 64k | 128k |
|---|---|---|---|---|---|---|---|---|
| Independent | 0.00 | 0.00 | 0.00 | 0.00 | 0.01 | 0.01 | 0.03 | 0.06 |
| PrivBayes | 0.00 | 0.00 | 0.00 | 0.00 | 0.01 | 0.01 | 0.03 | 0.05 |
| MST | 0.01 | 0.01 | 0.01 | 0.01 | 0.02 | 0.02 | 0.04 | 0.07 |
| DP-WGAN | 0.00 | 0.00 | 0.00 | 0.00 | 0.01 | 0.01 | 0.02 | 0.04 |
| PATE-GAN | 0.00 | 0.00 | 0.00 | 0.00 | 0.00 | 0.00 | 0.00 | 0.01 |

**Table 5:** Running time (in minutes) of the model's generation step for fitted DP generative models, on *Corr Gauss*, varying $n$ and $d = 32$.

- The first two columns of *Mix Gauss Unsup* (inspired by (Frigerio et al., 2019)) are a mixture of six gaussians distributed in a ring with center 0, while the remaining columns represent noise in the form of uncorrelated gaussians with mean 0. The model should be able to separate signal from noise and reproduce the six clusters.

- The *Mix Gauss Sup* dataset is the same as the previous one but with an added target column, labeling the six gaussians in a non-linearly separable way. Classifiers trained on the real and on the synthetic data should have similar performance.

For all datasets, we vary the number of columns in the range {8, 16, 32, 64, 128, 256, 512, 1,024} while keeping the rows to 16k. We also vary the number of rows in the range {250, 500, 1k, 2k, 4k, 8k, 16k, 32k, 64k, 128k} while fixing the columns to 32. When applicable, we create test datasets with size equal to 20% of the training.

**MNIST.** The MNIST dataset (LeCun et al., 2010) is a collection of greyscale handwritten digits. There are 60K training and 10K testing samples. The task is to classify the digit. We experiment with a varying number of features. On top of the original 28x28 pixels, we also rescale the images to 10x10, 16x16, and 22x22.

**Census.** The Census dataset (Dua & Graff, 2017) is extracted from the 1994 and 1995 Current Population Surveys conducted by the US Census Bureau. It contains 41 (six numerical and 35 categorical) demographic and employment-related variables. The target column indicates whether the individual's income exceeds $50k/year. The dataset consists of 199,523 training and 99,762 testing instances. We vary the training points in the range {1k, 2k, 4k, 8k, 16k, 32k, 64k, 128k, 199,523} (the latter being the original size).

## C    ADDITIONAL RESULTS AND PLOTS

### C.1    SCALABILITY

We report summaries of the running time of the fitting step for *Corr Gauss* with varying $n$ in Table 4 and generation step with varying $n$ and $d$ in Table 5, 6. The results are discussed in Sec. 4 (see the paragraph with heading *Scalability*).

### C.2    STATISTICS

We visualize the average statistics for *Eye Gauss* and *Corr Gauss* with varying dimensions; more precisely, mean and correlation for the former in Fig. 4, 5 and mean and correlation for the latter in Fig. 6, 7. The results are discussed in detail in Sec. 4.2.

| DP Model$\downarrow d\rightarrow$ | 8 | 16 | 32 | 64 | 128 | 256 | 512 | 1,024 |
|---|---|---|---|---|---|---|---|---|
| Independent | 0.00 | 0.00 | 0.01 | 0.02 | 0.07 | 0.24 | 0.99 | 3.69 |
| PrivBayes | 0.00 | 0.00 | 0.01 | 0.02 | 0.03 | 0.07 | | |
| MST | 0.00 | 0.01 | 0.02 | 0.05 | 0.13 | | | |
| DP-WGAN | 0.00 | 0.00 | 0.01 | 0.01 | 0.02 | 0.06 | 0.39 | 1.63 |
| PATE-GAN | 0.00 | 0.00 | 0.00 | 0.00 | 0.01 | 0.05 | 0.3 | 1.57 |

**Table 6:** Running time (in minutes) of the model's generation step for fitted DP generative models, on *Corr Gauss*, varying $d$ and $n = 16k$.

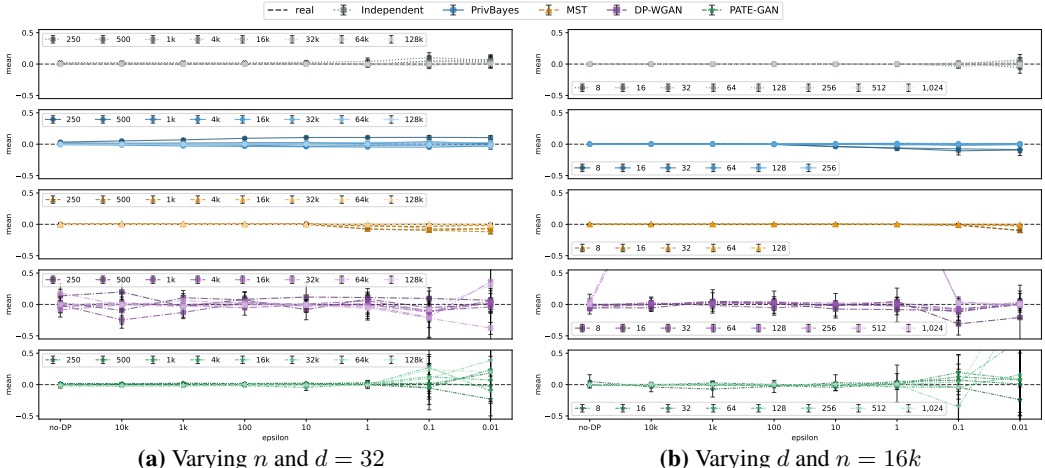

(a) Varying $n$ and $d = 32$    (b) Varying $d$ and $n = 16k$

**Figure 4:** Marginal mean for different $\epsilon$ levels, on *Eye Gauss*, varying $n$ and $d$.

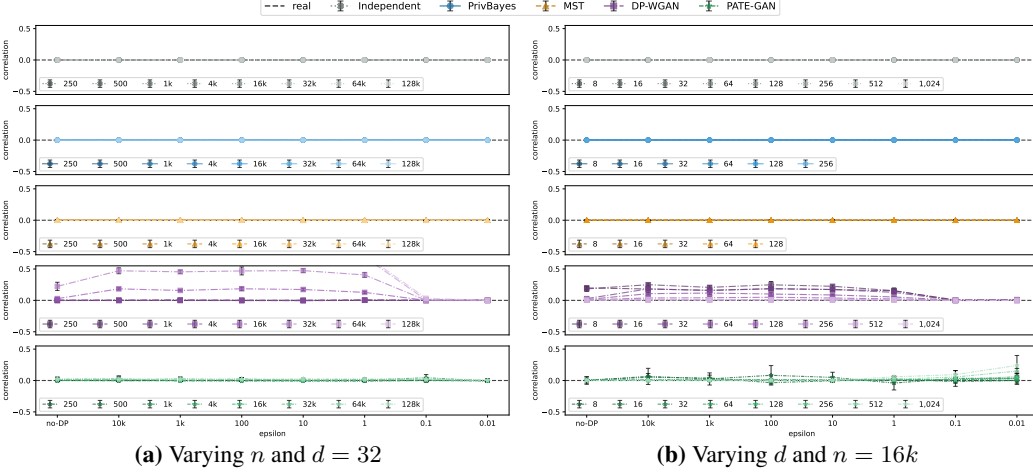

(a) Varying $n$ and $d = 32$    (b) Varying $d$ and $n = 16k$

**Figure 5:** Other (apart from diagonal) pairwise correlation for different $\epsilon$ levels, on *Eye Gauss*, varying $n$ and $d$.

### C.3 SIMILARITY

We plot zoomed-in average marginal and pairwise mutual information similarly for MST and PATE-GAN on *Census* with varying $n$ in Fig. 8. We also visualize fitted networks for MST and PrivBayes in Fig. 9 and break down the mutual information similarly between connected and non-connected nodes in Fig. 10. These experiments are discussed in Sec. 4.3.

### C.4 PCA AND CLUSTERING

We now focus on Principal Component Analysis (PCA) and clustering tasks.

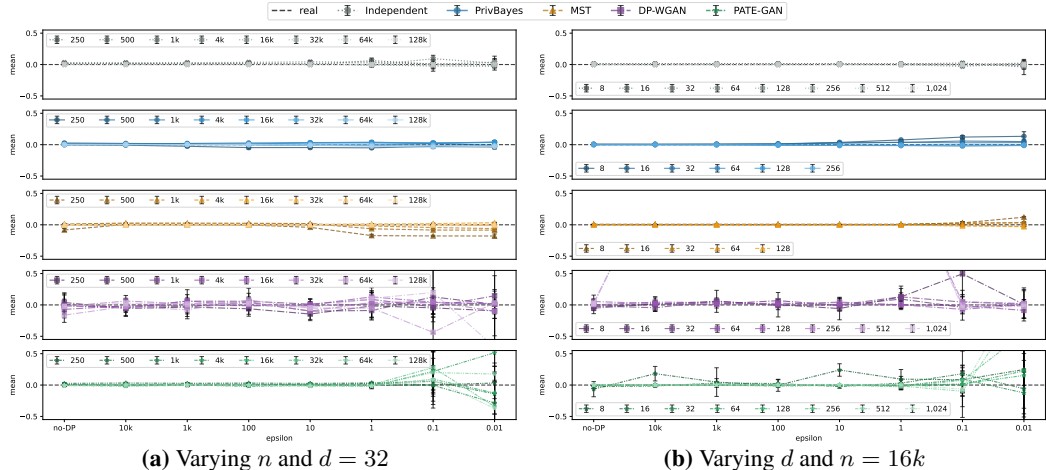

**(a)** Varying $n$ and $d = 32$

**(b)** Varying $d$ and $n = 16k$

**Figure 6:** Marginal mean for different $\epsilon$ levels, on *Corr Gauss*, varying $n$ and $d$.

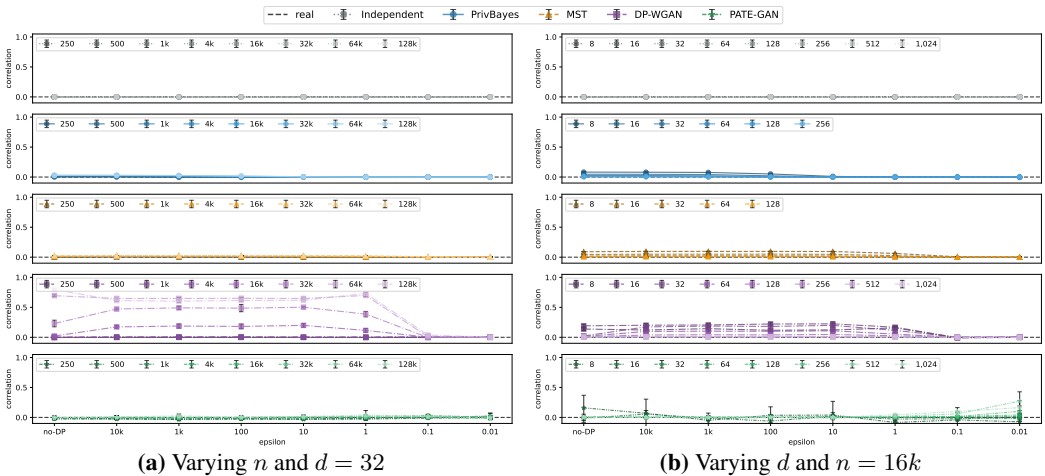

**(a)** Varying $n$ and $d = 32$

**(b)** Varying $d$ and $n = 16k$

**Figure 7:** Other (apart from diagonal and off-diagonal) pairwise correlation for different $\epsilon$ levels, on *Corr Gauss*, varying $n$ and $d$.

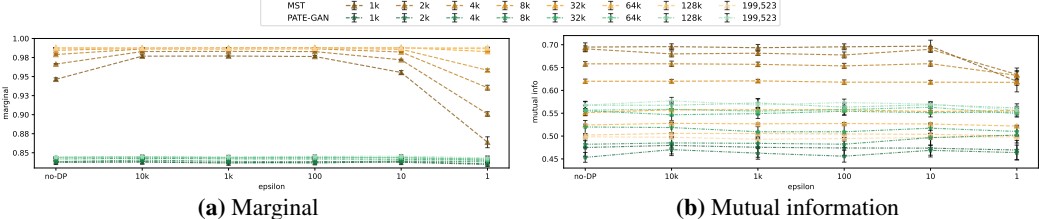

**(a)** Marginal

**(b)** Mutual information

**Figure 8:** Marginal and pairwise mutual information similarity zoomed in for MST and PATE-GAN for different $\epsilon$ levels, on *Census*, varying $n$.

The Kernel Density Estimation (KDE) on the first 2 PCA components of all four *Gauss* datasets with varying dimensions are plotted in Fig. 12, 13, 14, 15, 16, 17, 18, and 19 while the silhouette scores of clusters predicted by Mixture of Gaussians models fitted to the first 2 PCA components on *Corr Gauss* are shown in Fig. 11.

Analyzing the PCA plots, no models manages to replicate all datasets sufficiently well. PrivBayes probably comes closest for $\epsilon \geq 10$, and for the *Mix Gauss* datasets, it is the only model that properly separates the first two columns creating the six clusters from the rest containing noise. When a tighter privacy budget is applied, or the dataset dimensions are increased, the variance of the syn-

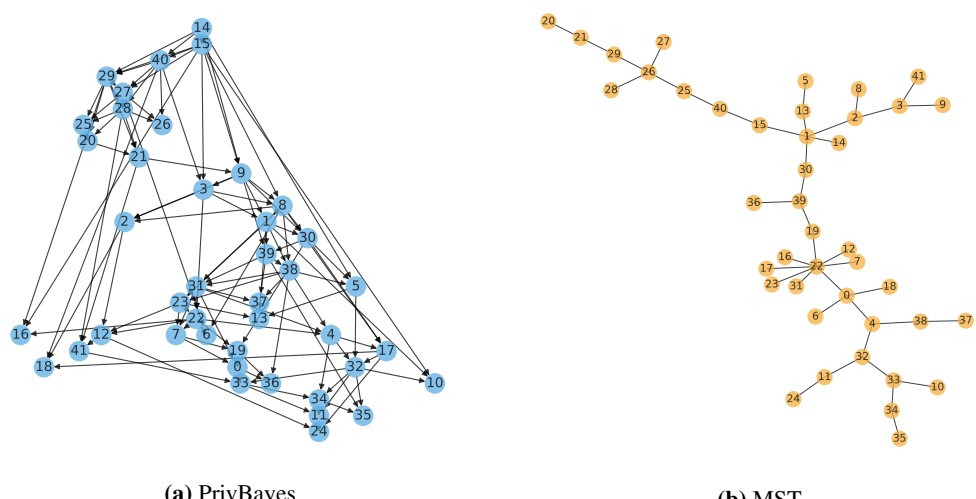

**(a)** PrivBayes

**(b)** MST

**Figure 9:** Example fitted networks for PrivBayes and MST with $\epsilon = 1$, on *Census*.

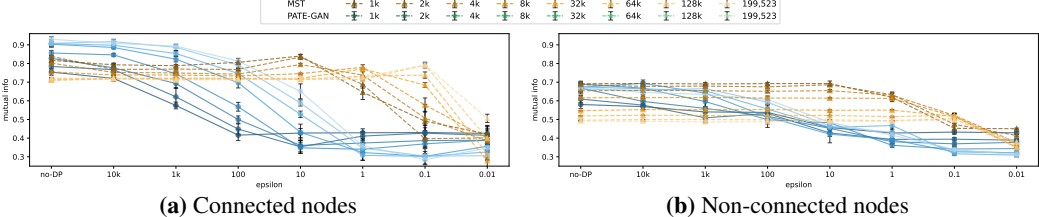

**(a)** Connected nodes

**(b)** Non-connected nodes

**Figure 10:** Pairwise mutual information (connected/non-connected nodes extracted from the fitted networks for PrivBayes and MST) similarity for different $\epsilon$ levels, on *Census*, varying $n$.

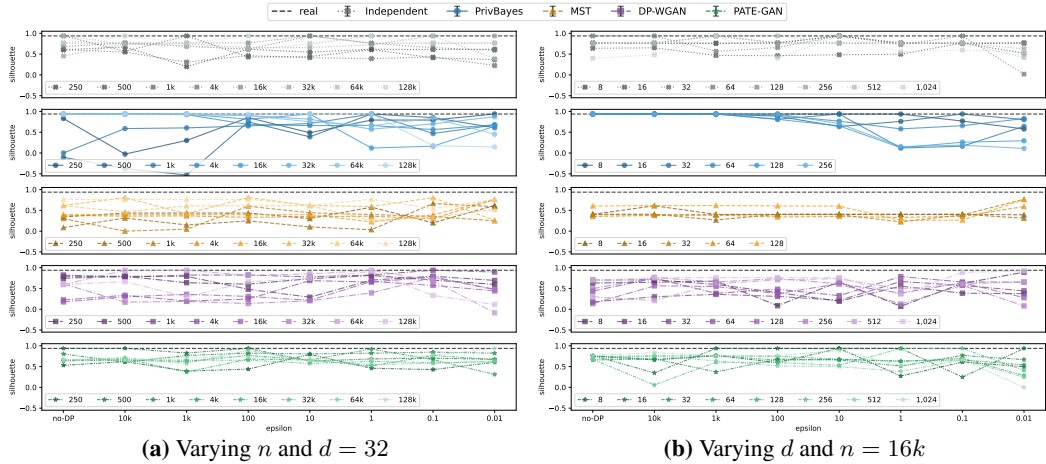

**(a)** Varying $n$ and $d = 32$

**(b)** Varying $d$ and $n = 16k$

**Figure 11:** Silhouette score for different $\epsilon$ levels, on *Mix Gauss Unsup*, varying $n$ and $d$.

thetic data increases too for all datasets. While MST performs very well for eye and *Corr Gauss*, it ultimately fails for the other two as it generates some difficult-to-define structure, probably due to mode collapse. Unsurprisingly, Independent cannot capture *Mix Gauss* distributions either. Unfortunately, the GAN models do not perform well on these datasets either. At the very least, they capture the range of the data well. However, $n$ and $d$ do not seem to be important factors. For $d \geq 32$ and $\epsilon > 0.1$, PATE-GAN generates data that loosely resembles the real, but it is shaped more as a box rather than a ring.

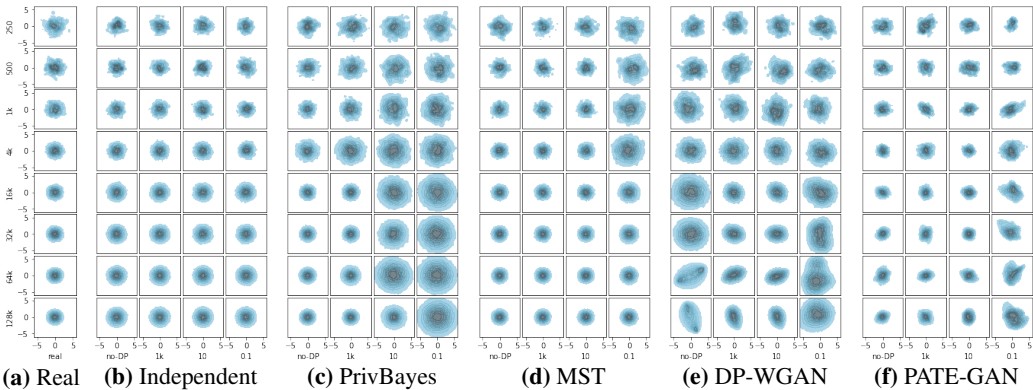

**(a)** Real  **(b)** Independent  **(c)** PrivBayes  **(d)** MST  **(e)** DP-WGAN  **(f)** PATE-GAN

**Figure 12:** KDE on the first 2 PCA principles for different $\epsilon$ levels, on *Eye Gauss*, varying $n$.

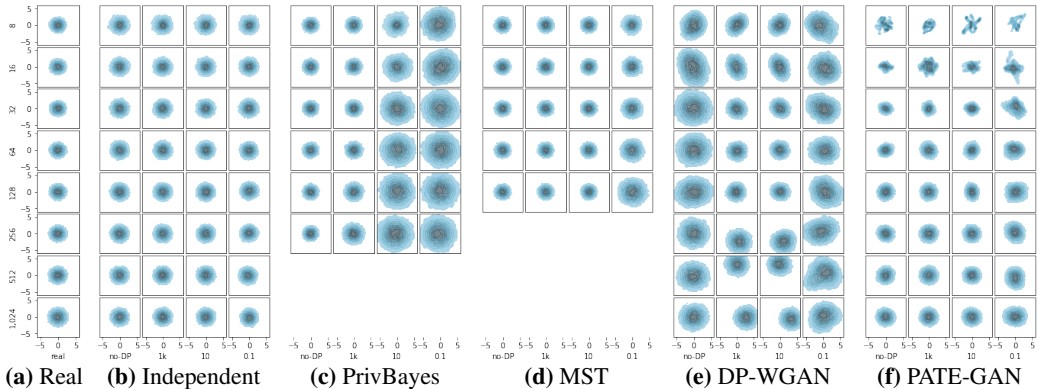

**(a)** Real  **(b)** Independent  **(c)** PrivBayes  **(d)** MST  **(e)** DP-WGAN  **(f)** PATE-GAN

**Figure 13:** KDE on the first 2 PCA principles for different $\epsilon$ levels, on *Eye Gauss*, varying $d$.

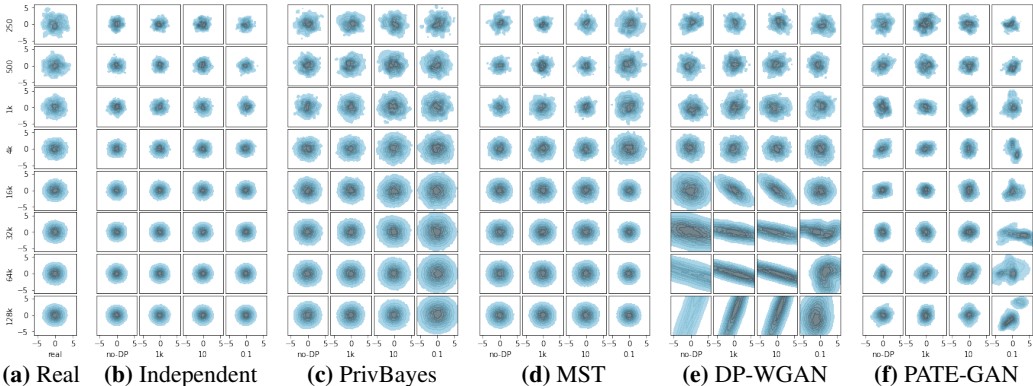

**(a)** Real  **(b)** Independent  **(c)** PrivBayes  **(d)** MST  **(e)** DP-WGAN  **(f)** PATE-GAN

**Figure 14:** KDE on the first 2 PCA principles for different $\epsilon$ levels, on *Corr Gauss*, varying $n$.

The silhouette scores in Fig. 11 look very variably and noisy, most likely because the clustering was done on the first 2 PCA components as opposed to the complete datasets due to computational time restrictions. Broadly speaking, they confirm what we observed in the PCA plots. PrivBayes performs very well for $n > 1k$. Varying $n$ has a strong effect on MST and makes the scores very noisy and unpredictable. PATE-GAN achieves good silhouette scores, greater or equal to 0.45 for all $n$.

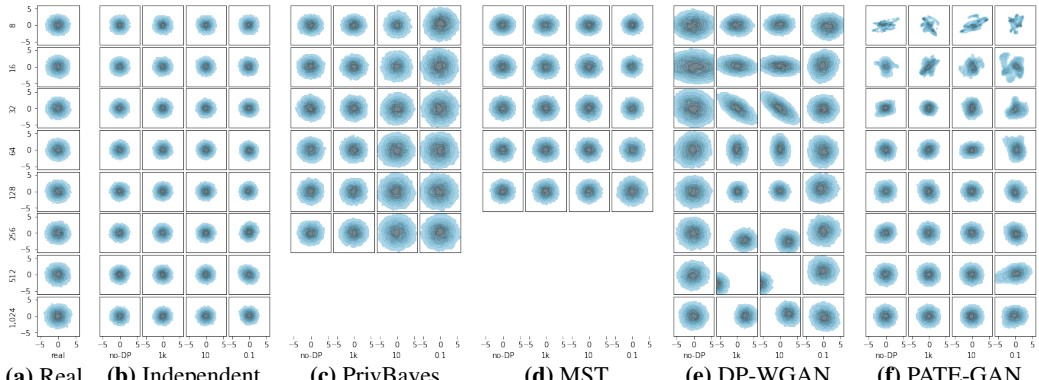

**(a)** Real **(b)** Independent **(c)** PrivBayes **(d)** MST **(e)** DP-WGAN **(f)** PATE-GAN

**Figure 15:** KDE on the first 2 PCA principles for different $\epsilon$ levels, on *Corr Gauss*, varying $d$.

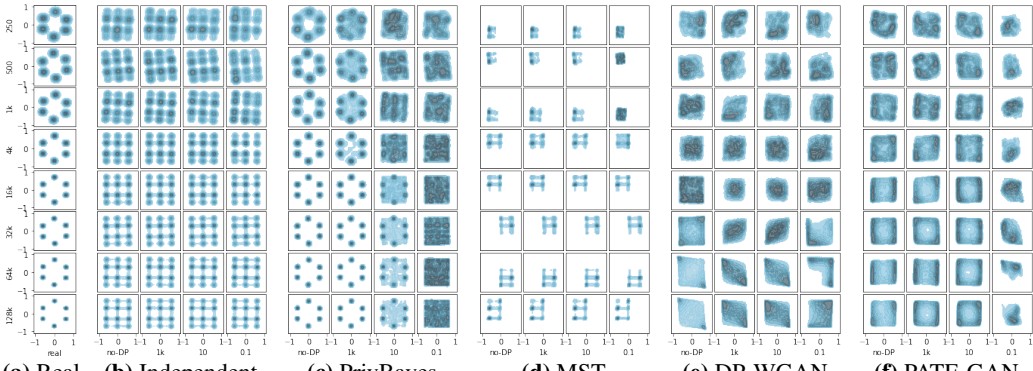

**(a)** Real **(b)** Independent **(c)** PrivBayes **(d)** MST **(e)** DP-WGAN **(f)** PATE-GAN

**Figure 16:** KDE on the first 2 PCA principles for different $\epsilon$ levels, on *Mix Gauss Unsup*, varying $n$.

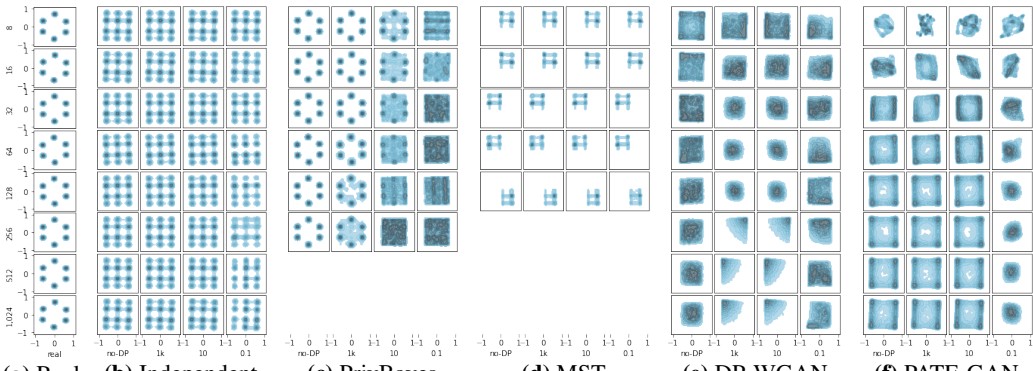

**(a)** Real **(b)** Independent **(c)** PrivBayes **(d)** MST **(e)** DP-WGAN **(f)** PATE-GAN

**Figure 17:** KDE on the first 2 PCA principles for different $\epsilon$ levels, on *Mix Gauss Unsup*, varying $d$.

## C.5 CLASSIFICATION

The average accuracy for *Mix Gauss Sup* with varying dimensions is plotted in Fig. 20 while the average accuracy and F1 for *Census* with increasing $n$ is displayed in Fig. 21. We look at these plots in detail in Sec. 4.4.

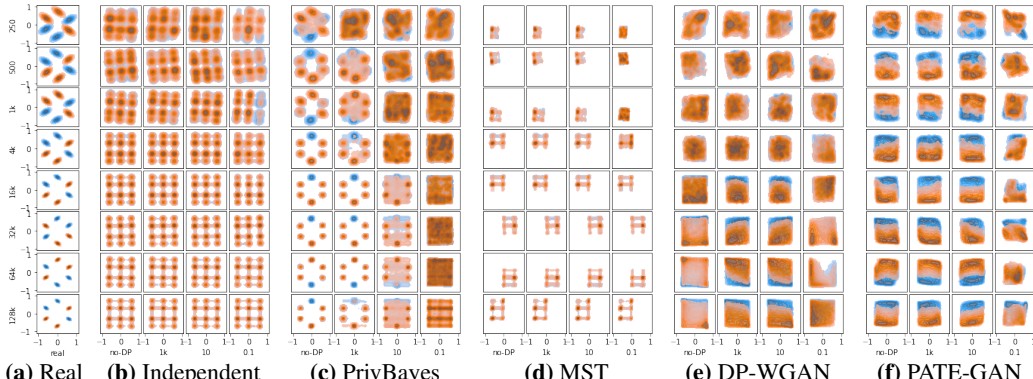

**(a)** Real  **(b)** Independent  **(c)** PrivBayes  **(d)** MST  **(e)** DP-WGAN  **(f)** PATE-GAN

**Figure 18:** KDE on the first 2 PCA principles for different $\epsilon$ levels, on *Mix Gauss Sup*, varying $n$.

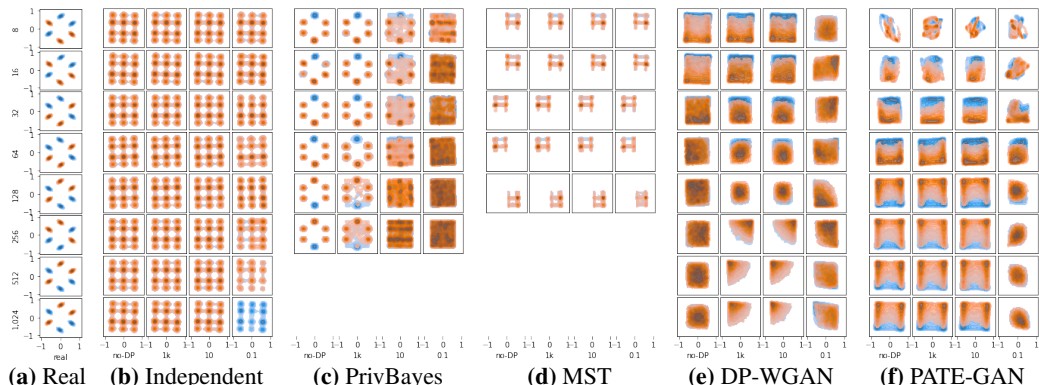

**(a)** Real  **(b)** Independent  **(c)** PrivBayes  **(d)** MST  **(e)** DP-WGAN  **(f)** PATE-GAN

**Figure 19:** KDE on the first 2 PCA principles for different $\epsilon$ levels, on *Mix Gauss Sup*, varying $d$.

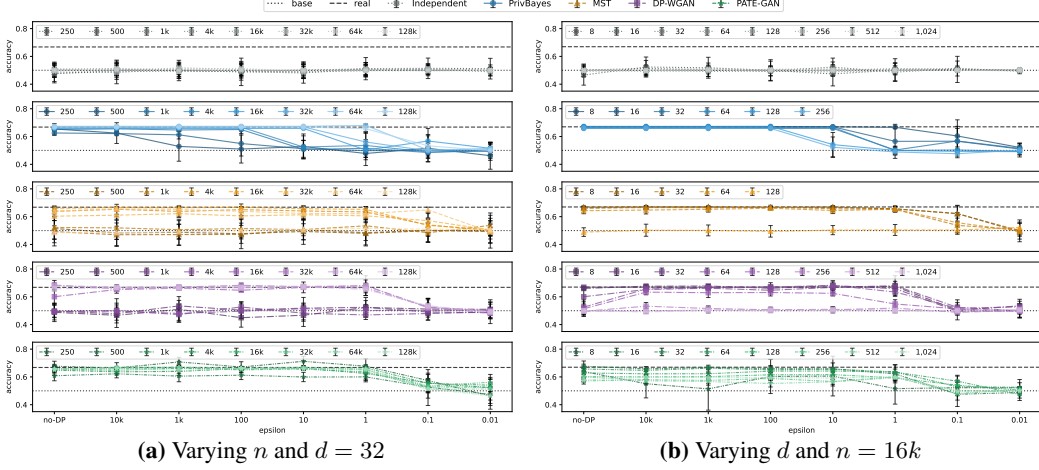

**(a)** Varying $n$ and $d = 32$  **(b)** Varying $d$ and $n = 16k$

**Figure 20:** Accuracy for different $\epsilon$ levels, on *Mix Gauss Sup*, varying $n$ and $d$.

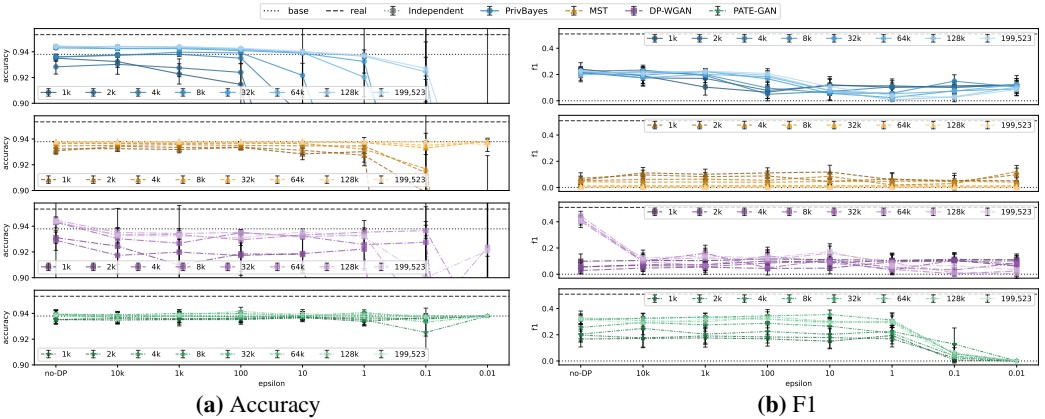

**(a)** Accuracy

**(b)** F1

**Figure 21:** Accuracy and F1 for different $\epsilon$ levels, on *Census*, varying $n$.

