# OpenReview forum: "Mind the Privacy Budget: How Generative Models Spend their Privacy Budgets"
_ICLR.cc/2023/Conference — Submitted to ICLR 2023_

### Official Review · Reviewer_uTaB · 2022-10-19

**Confidence:** 4
**Correctness:** 4
**Technical Novelty And Significance:** 2
**Empirical Novelty And Significance:** 2
**Recommendation:** 5

**Clarity, Quality, Novelty And Reproducibility:**

clarity: the paper has clear writing.
quality: the paper designed careful experiments to benchmark performance, but lacks more detailed ablation studies to uncover reasons why certain methods behave in certain ways.
novelty: some of the empirical observations are novel. but overall, the paper benchmarking existing methods and most observations are expected.
repro: results seem reproducible, to the best of my knowledge.

**Strength And Weaknesses:**

strength: the paper is empirical in natural and aims to answer concrete practical questions. in this regard, the paper scores nicely as the experiments on evaluating performance is somewhat comprehensive in my opinion.

weakness: i see two major drawbacks of this work, despite of its wide-covering experiments.

1) many discoveries in this paper are rather intuitive and follows naturally from inspecting the algorithms. e.g., "PrivBayes' performance on downstream tasks degrades when a stricter privacy budget is imposed..." is the expected and much empirical verification is given in the original PrivBayes paper [1]. in this regard, there are only few truly novel empirical observations uncovered in the current work.

2) the work uncovers two (somewhat) unexpected results in my opinion, but falls short in helping us understand how these empirical phenomena arise. e.g., on page 9, the paper suggests that "we observed that more data does not always translate to improved quality for all models and evaluation." this is quite unexpected for DP machine learning, as workflows as such tend to be bottlenecked by fitting the training data (as opposed to combating overfitting). it'd therefore be useful to design and run experiments to understand why such observations are the case. for instance, are the results due to experimental noise? are the non-monotone results w/ GANs due to bad optimization? understanding why certain counter-intuitive behavior emerges would help future research in developing better methods.


[1] Zhang, Jun, et al. "Privbayes: Private data release via bayesian networks." ACM Transactions on Database Systems (TODS) 42.4 (2017): 1-41.

**Summary Of The Paper:**

the paper benchmarks how various existing approaches for the problem of *DP synthetic tabular data generation* perform as the number of features and instances of the dataset change. in particular, the paper answers some previous under-studied research questions such as 1) how do methods scale with the number of columns.

**Summary Of The Review:**

the paper designed careful experiments to benchmark performance, but lacks more detailed ablation studies to uncover reasons why certain methods behave in certain ways.

---

> ### Author Response · Authors · 2022-11-16
> **Author Response to Reviewer uTaB**
>
> We are glad the reviewer finds our experiments comprehensive and we would like to thank them for their time, raised questions, and offered suggestions. We now answer specific comments.
>
>
> > many discoveries in this paper are rather intuitive and follows naturally from inspecting the algorithms…. in this regard, there are only few truly novel empirical observations uncovered in the current work.
>
> We agree with the reviewer that simpler and more studied models (such as PrivBayes) display a more natural behavior (i.e., close to initial expectations). However, we also showcase a number of novel contributions:
> * We consistently demonstrate that MST is not the best model in all settings (which contradicts the established narrative [1,2,3]); Also, we find out that PrivBayes is actually more scalable than MST.
> * We highlight in which settings (higher dimensions and more complex tasks) GANs are highly competitive and useful and why (this again contradicts studies in tabular data  [1, 3]).
> * We show that applying a small degree of privacy could actually serve as a regularizer and help the performance of the model when there is not enough training data; this was previously unobserved for DP tabular data
> * We consistently observe that more training data does not necessarily translate to better performance (which, once again, contradicts the common assumption).
>
> Overall, we are confident that our contributions will assist researchers and practitioners deploying DP synthetic data techniques in understanding the trade-offs and navigating through the best candidate models.
>
>
> > the work uncovers two (somewhat) unexpected results in my opinion, but falls short in helping us understand how these empirical phenomena arise.
>
> We believe we have run enough experiments to find the source or main contributing factor for the findings. Next, we go over them in the same order:
> * The underlying factor is that MST is overfitting to the objective function (i.e., 2/3-way marginals), which actually worsens the performance on the downstream task (if different from the objective function) when trained on more data (discussed in 4.3 and 4.4).
> * Again due to the objective function, GANs prioritize creating plausible synthetic data points (i.e., implicitly maintaining correlations between columns), rather than maintaining overall dataset statistics (discussed in 4.3).
> * Concretely for MST (which applies the Gaussian mechanisms), this is only true when the task is relatively simple (i.e., capturing correlation/marginal similarity) and there is not enough training data (discussed in 4.2 and 4.3).
> * We analyze these scenarios and find out that this is due to the objective function (MST) or the unpredictability of the model (GANs) (discussed in 4.2 and 4.4).
>
> We thank the reviewer again for their thoughtful suggestions, we would like to ask for further advice:
> * Apart from the experiments we have devised to explain the models’ behaviors, what further experiments does the reviewer think would enhance our explanations?
> * Is there any particular set of tasks or further experiments that the reviewer believes will lead to new/unexpected findings?
>
>
>
> [1] NIST. 2018 Differential Privacy Synthetic Data Challenge. https://www.nist.gov/ctl/pscr/ open-innovation-prize-challenges/past-prize-challenges/2018-differential-privacy-synthetic, 2018.
>
> [2] Ryan McKenna, Terrance Liu. A simple recipe for private synthetic data generation, https://differentialprivacy.org/synth-data-1/, 2022
>
> [3] Yuchao Tao, Ryan McKenna, Michael Hay, Ashwin Machanavajjhala, and Gerome Miklau. Benchmarking differentially private synthetic data generation algorithms. AAAI Workshop on Privacy-Preserving Artificial Intelligence (PPAI), 2022.

---

### Official Review · Reviewer_Rx8R · 2022-10-24

**Confidence:** 5
**Correctness:** 4
**Technical Novelty And Significance:** 1
**Empirical Novelty And Significance:** 2
**Recommendation:** 3

**Clarity, Quality, Novelty And Reproducibility:**

While the paper is generally well written and easy to follow, the figures are uniformly too small with too much information compressed into them.

**Strength And Weaknesses:**

Strengths:
1) The experimental setup used by the authors is very rigorous
2) The question of how different generative models differ as a function of N, d and epsilon are somewhat interesting questions.

Weakness:
1) Beyond highlighting some of the differences between the 4 generative modeling approaches, I am struggling to find the novel contribution of the paper.
2) The title of the paper and the claim that it examines how the privacy budget is being utilized is quite misleading in my opinion.
3) The authors only consider models for tabular data generation here but the paper could be significantly improved by considering models that generate more complex data modalities e.g. images. In the case of images for instance, one might be interested in wondering how the difference in learned convolutional filters as a function of N, d and epsilon.

**Summary Of The Paper:**

In this paper, the authors rigorously study the effects of varying N (number of samples) and d (dimensionality of the dataset) for 4 separate private generative modeling approaches (belonging to two different classes of algorithms). This is done at different privacy levels (epsilon) and the evaluation is based on both generative modeling metrics (correlation/mutual information between variables) as well as classification performance. As a result of this exercise, the authors are able to report some unique behaviors of different private generative modeling approaches.

**Summary Of The Review:**

While I like the rigor of the evaluations done by the authors, I believe there aren't very many novel generalizable insights that come from the current version of the paper. As such, it reads more like a workshop paper at the moment. However, if the authors were to expand the paper to models for more complex data modalities, I believe it might make for an interesting conference paper.

---

> ### Author Response · Authors · 2022-11-16
> **Author Response to Reviewer Rx8R**
>
> We thank the reviewer for their time and comments. Next, we respond to specific points raised in the review.
>
>
> > Beyond highlighting some of the differences between the 4 generative modeling approaches, I am struggling to find the novel contribution of the paper.
>
> Apart from highlighting the difference between the 4 models (in terms of model architectures), we believe that that our paper has a number of novel contributions (that have all been understudied):
> * We analyze the different DP mechanisms (every model utilizes a different DP mechanism) and their effect on the downstream task with varying dataset dimensions (in 3.2); then, we run numerous experiments in 4.
> * We consistently demonstrate that MST is not the best model in all settings (which contradicts the established narrative [1,2,3]); furthermore, we analyze the underlying factor – overfitting to the objective function, which actually worsens the performance on the downstream task when trained on more data. Also, we find out that PrivBayes is actually more scalable than MST.
> * We highlight in which settings (higher dimensions and more complex tasks) GANs are highly competitive and useful and why (this again contradicts studies in tabular data [1,3]).
> * We show that applying a small degree of privacy could actually serve as a regulazier and help the performance of the model when there is not enough training data; this was previously unobserved for DP tabular data
> * We consistently observe that more training data does not necessarily translate to better performance (which again contradicts the common assumption). We also analyze these scenarios and find out that this could be due to the objective function (MST) or the unpredictability of the model (GANs).
>
> Overall, we are confident that our contributions will assist researchers and practitioners deploying DP synthetic data techniques in understanding the trade-offs and navigating through the best candidate models.
>
>
> > The title of the paper and the claim that it examines how the privacy budget is being utilized is quite misleading in my opinion.
>
> We believe that most generative models in literature are built with having the available/proven DP mechanisms satisfying DP in mind (this is definitely true for all 4 models presented in our paper). Making a decision on the DP mechanism is one of most important architectural choices which determines exactly how the privacy budget would be distributed, thus being a major factor for all downstream tasks. Furthermore, in our paper we study 4 models that all utilize different DP mechanisms, we analyze them in 3.2, and measure the effect they have throughout 4.
>
>
> > The authors only consider models for tabular data generation here but the paper could be significantly improved by considering models that generate more complex data modalities e.g. images. In the case of images for instance, one might be interested in wondering how the difference in learned convolutional filters as a function of N, d and epsilon.
>
> We agree, our main focus is tabular data since all 4 generative models have been proposed for this domain. While studying CNNs and the number of convolutions in the context of DP sounds really interesting, it should be considered outside of the scope of this work as this analysis could be applied to only a portion of the models (GANs) as the rest of the models cannot deal with high-dimensional data like images.
>
>
> We thank the reviewer again for their helpful suggestions, we would like to ask for further advice:
> * Beyond images (since we would like to keep the main focus tabular data) what specific modalities that may be more complex would you recommend?
>
>
>
> [1] NIST. 2018 Differential Privacy Synthetic Data Challenge. https://www.nist.gov/ctl/pscr/ open-innovation-prize-challenges/past-prize-challenges/2018-differential-privacy-synthetic, 2018.
>
> [2] Ryan McKenna, Terrance Liu. A simple recipe for private synthetic data generation, https://differentialprivacy.org/synth-data-1/, 2022
>
> [3] Yuchao Tao, Ryan McKenna, Michael Hay, Ashwin Machanavajjhala, and Gerome Miklau. Benchmarking differentially private synthetic data generation algorithms. AAAI Workshop on Privacy-Preserving Artificial Intelligence (PPAI), 2022.

---

### Official Review · Reviewer_vrg9 · 2022-10-31

**Confidence:** 4
**Correctness:** 4
**Technical Novelty And Significance:** 2
**Empirical Novelty And Significance:** 3
**Recommendation:** 5

**Clarity, Quality, Novelty And Reproducibility:**

* I think the evaluation is decent but could be expanded to a wider set of tasks, including a wider set of tabular datasets, e.g., the entirety of the UCI ML repository.
* The work is fairly original to my knowledge, it presents a wide empirical evaluation.
* The writing is overall somewhat clear, but it is a bit repetitive between Sections 1, 4.5, and 5. It also spends a lot of the space introducing all of the methods compared.

**Strength And Weaknesses:**

Strengths:
* This is very much a "in-the-weeds" paper that compares various methods on different metrics and datasets. I think some of the resulting findings will be useful for practicioners, e.g., runtime of various methods as you vary the number of features and number of rows. I also think for researchers developing new DP generative models, it'd be useful for them to evaluate their models on metrics like the ones proposed here.
* I find it interesting that there are cases when using more training data can sometimes hurt. It adds to the growing literature that tuning hyperparameters when training with DP can be quite difficult, non-intuitive, and not in line with what works best for non-private ML.
* I like seeing downstream comparison of methods from different model classes, e.g., deep generative models trained with DP-SGD versus graphical modeling approaches.

Weaknesses:
* One concern I have is that some of the "main findings" of the paper are actually just statements about the properties of the methods themselves (see introduction and RQ1/RQ2 in Section 4.5). For example, the claim that "deep generative models spend their budget per training iteration and can handle much wider datasets but become slower with more data" can be made without an empirical evaluation---the method has those strengths/weaknesses because of the very nature of how DP-SGD and neural nets work.
* While I think tabular datasets are important to study for a wide variety of real-world settings, 4/6 of the tasks in this paper are variants of sampling from a gaussian distribution.

**Summary Of The Paper:**

Synthetic data generation is a key application of differentially-private generative models---it allows you to build a synthetic dataset that can be used repeatedly without privacy risks for downstream ML or data science use cases. This paper presents an empirical evaluation of different DP generative models on mainly tabular datasets, e.g., studying how well they recreate the underlying data distribution and the various training costs.

**Summary Of The Review:**

I think some of the findings can be useful to a broader community, but overall I find the evaluation and takeaways to be too limited to warrant a publication.

---

> ### Author Response · Authors · 2022-11-16
> **Author Response to Reviewer vrg9**
>
> We thank the reviewer for finding our work interesting and useful. We also thank them for their time and thoughtful comments/suggestions. We answer specific points below, followed up by some questions.
>
>
>
>
> > One concern I have is that some of the "main findings" of the paper are actually just statements about the properties of the methods themselves (see introduction and RQ1/RQ2 in Section 4.5). For example, the claim that "deep generative models spend their budget per training iteration and can handle much wider datasets but become slower with more data" can be made without an empirical evaluation---the method has those strengths/weaknesses because of the very nature of how DP-SGD and neural nets work.
>
> We agree that most of these statements can be made without an empirical evaluation (e.g., most of the findings from RQ1/RQ2). Due to this, we spend 3.2 analyzing what DP mechanisms the different models use and how that could affect their performance without running any experiments. However, it is not clear in advance which of the two graphical models (PrivBayes vs MST) would be faster/less computationally expensive. This is also the case when comparing DP-WGAN vs PATE-GAN. We would like to point out that we condense the findings from RQ1/RQ2 in Main Finding #1 in Introduction.
> Overall, in our mind, both RQ1 and RQ2 should be considered as interim conclusions that set the scene for RQ3. In fact, having answered RQ1 and RQ2, the majority of our work focuses on evaluating the effects on various downstream tasks (leading to Main Finding #2, 3, and 4 in Introduction and the overwhelming majority of 4).
>
>
> > While I think tabular datasets are important to study for a wide variety of real-world settings, 4/6 of the tasks in this paper are variants of sampling from a gaussian distribution.
>
> Using the 4 gaussian datasets with varying levels of complexity is pivotal to profile the behavior of the evaluated models (something which hasn’t been done before in such rigor) and provides more insights than just numbers. Moreover, the controlled datasets allow us to very clearly identify failure modes (e.g., apart from PrivBayes, all models fail in separating signal from noise in the clustering task; GANs have very noisy performance on simple statistics capturing tasks), which is an understudied problem in the context of DP synthetic data.
>
>
> > I think the evaluation is decent but could be expanded to a wider set of tasks, including a wider set of tabular datasets, e.g., the entirety of the UCI ML repository.
>
> Ideally, we would like to run an even larger set of experiments. However, we would like to reiterate that these experiments are costly (both in terms of time and computation). For this paper, we have trained around 15,000 generative models and generated 75,000 synthetic datasets. Our findings are consistent across the various datasets, thus,  we believe that running even more experiments is unlikely to result in new findings.
>
>
> We thank the reviewer again for their helpful suggestions, we would like to ask for further advice:
> * Is there any particular set of tasks or tabular datasets that the reviewer believes will lead to new/unexpected findings (apart from these already presented in our work)?

---

### Decision · Program_Chairs · 2023-01-20

**Decision:**

Reject

**Justification For Why Not Higher Score:**

More exploration of counterintuitive phenomenon needed.

**Justification For Why Not Lower Score:**

N/A

**Metareview: Summary, Strengths And Weaknesses:**

The reviewers felt the paper had some solid experimental exploration. The critiques centered largely around low novelty and that the main findings were just confirmations of known or intuitive facts. The most interesting findings were those which were highly counterintuitive, such as more data actually hurting. However, there was too little exploration/explanation of these phenomena. The authors are suggested to revise to focus more on exploring new and counterintuitive results, and perhaps omitting the straightforward results.